# Perfect linear optics using silicon photonics

Miltiadis Moralis-Pegios [1,3] ✉, George Giamougiannis [1,3], Apostolos Tsakyridis[1], David Lazovsky[2] & Nikos Pleros[1]

Recently there has been growing interest in using photonics to perform the linear algebra operations of neuromorphic and quantum computing applications, aiming at harnessing silicon photonics' (SiPho) high-speed and energy-efficiency credentials. Accurately mapping, however, a matrix into optics remains challenging, since state-of-the-art optical architectures are sensitive to fabrication imperfections. This leads to reduced fidelity that degrades as the insertion losses of the optical matrix nodes or the matrix dimensions increase. In this work, we present the experimental deployment of a 4 × 4 coherent crossbar (Xbar) as a silicon chip and validate experimentally its theoretically predicted fidelity restoration credentials. We demonstrate the experimental implementation of 10,000 arbitrary linear transformations achieving a record-high fidelity of 99.997% ± 0.002, limited mainly by the measurement equipment. Our work represents an integrated optical circuit providing almost unity and loss-independent fidelity in the realization of arbitrary matrices, highlighting light's credentials in resolving complex computations.

Linear optical circuitry has been recently brought into the spotlight as a powerful tool for a wide range of applications, including microwave and quantum optical signal processing[1,2], as well as next-generation bio-inspired computing architectures. Recent architectural breakthroughs[3–9] together with the growing maturity of SiPho platforms[10], which allows the deployment of hundreds to thousands of miniaturized photonic components into a single chip, has elevated integrated photonics to the platform of choice for next generation linear optical circuitry, with recent experimental demonstrations confirming the potential of silicon photonics to break through the stability, scale and energy consumption barriers of alternative platforms, such as bulk optics[11].

The dominant approach in integrated linear photonic circuitry architectures is based until now on manipulating light via coherent interferometric meshes[12–14] that target the implementation of unitary linear transformations. These layouts rely on U(2)-based matrix decomposition schemes and cascaded stages of 2 × 2 Mach-Zehnder interferometer (MZI) blocks. Despite their simplicity, symmetrical layout and impressive academic and industrial demonstrations[15,16], these approaches are burdened by inherent architectural deficiencies that mainly originate from the accumulating nature of fabrication imperfections across their cascaded nodes. This has a significant impact to their size scaling and overall performance credentials[14] i.e., fidelity, defined as the Frobenious inner product of the targeted and the experimentally obtained output vectors, programming complexity, processing bandwidth, energy- and footprint-efficiency. More specifically, the hardware implementation of such meshes suffers from (i) insertion loss (IL) that scales nonlinearly to the loss of the constituent matrix node, limiting in this way the deployment of large-size photonic processors, (ii) high-complexity programming[17,18], which becomes more pronounced when high-scale implementations are targeted, (iii) limited computing nodes' update speed, since cascading several high-speed and, as such, relatively high IL nodes, converges rapidly to prohibitive power budgets[14], (iv) reduced and non-restorable fidelity due to the intrinsic differential optical path losses, negating any perspective for perfectly mapping a targeted linear transformation into the optical domain, and finally (v) an application portfolio that is inherently confined to unitary transformations[12,13], with their universalization requiring the adoption of Singular Value Decomposition (SVD) schemes[19] with two unitary MZI meshes around a diagonal optical matrix, which pronounces all the above disadvantages[20]. Nevertheless, the advent of photonic integration technology has managed to yield ultra-low loss 2 × 2 MZI technology that allowed for significant progress in the scalability and IL

---

[1]Department of Informatics, Aristotle University of Thessaloniki, 54124 Thessaloniki, Greece. [2]Celestial AI, 2962 Bunker Hill Ln, Suite 200, Santa Clara, CA 95054, USA. [3]These authors contributed equally: Miltiadis Moralis-Pegios, George Giamougiannis. ✉e-mail: mmoralis@csd.auth.gr

performance of unitary and universal linear optical circuitry, although not being able yet to combine high-bandwidth operation with this ultra-low loss envelope. Programming complexity and mesh calibration has been also facilitated through self-configured linear optics[19]. However, fidelity restoration and inaccurate mapping of the linear transformations comprise native architectural drawbacks in these layouts; despite the fidelity improvements obtained through MZI node loss reductions, the inherent presence of differential paths cancels any perspective for restoring the circuit fidelity and accurately representing a targeted matrix into the optical experimental domain. On the contrary, in ring-based architectures[21–23], that harness the synthetic frequency dimension and/or wavelength division multiplexing (WDM) capabilities of light, programming comprises a straightforward procedure that can increase their computation robustness and fidelity performance. Yet, in these implementations the circuit complexity shifts to the unit required for generating multiple optical channels; this gets even more pronounced when targeting a fully-integrated system, where the need for multiple wavelength channels and as such resonant devices will inevitably impose significant scalability challenges, since the temperature variation would necessitate precise control of the ring resonances.

In this paper, we present perfect on-chip universal linear optics via a linear operator fabricated in Sipho platform, that is capable of breaking through the IL-fidelity-scale trade-offs of MZI-mesh based approaches, demonstrating experimentally perfect linear optical transformations even when high-loss optical matrix nodes are employed. This performance is enabled by fabricating a coherent photonic Crossbar (Xbar) architecture[14,24,25] onto the silicon platform, investing in a linear optical circuit architecture that demarcates from traditional unitary-based approaches and promotes a distributed tree-based power split-and-recombination stage together with a bijective weight mapping. The fabricated 4 × 4 Xbar employs 50 GHz silicon germanium (SiGe)-based electro-absorption modulators (EAMs) for the encoding of both the 4-element input vector and the 4 × 4 transformation matrix, using in addition silicon-based thermo-optic (TO) phase shifters (PSs) for fine phase control. A hardware aware (HA) calibration algorithm is proposed and employed for accurately assessing the performance of each individual EAM-based Xbar matrix cell, substantially decreasing the programming error compared to the baseline performance. Finally, we report an experimental record-high on-chip fidelity of 99.997 ± 0.002% in the implementation of 10,000 arbitrary linear transformations, a performance converging to the measurement uncertainty originating from the employed experimental equipment, highlighting both the fidelity restoration

capabilities of our architecture and the scalable and low-complexity programming required.

## Results

### Photonic Xbar architecture

The architecture of the proposed silicon photonic Xbar[14] is illustrated in Fig. 1a. As extensively analyzed in our previous work[14], an N × M Xbar operates as a vector-matrix linear operator via the direct mapping of the scalars of an N-elements-long vector X and an N × M matrix W into individual photonic modulating components. The Xbar architecture exploits the coherent recombination of modulated light via MZIs that are nested into splitting and recombining tree configurations. In particular, the N-elements of the input vector X are encoded to equivalent modulators located at the #N outputs of a 1:N splitter, with the resulting modulated light getting equally shared among the #M Xbar columns. A modulator at each row of each column weighs the respective element of X, according to the corresponding element of W. The weighted vector's elements of the #N rows of each column are summed via an N:1 combiner, completing the required linear transformation in the electric field. The vector and matrix element encoding sections are highlighted in the red and yellow rectangles of Fig. 1a, respectively.

Figure 1b depicts a microscope photo of the wirebonded 4 × 4 Xbar prototype, that was fabricated on imec's ISIPP50G platform using pdk-ready components. SiGe EAMs of 50 um length and a 3 dB bandwidth of 50 GHz[26] were deployed for the vector and matrix element encoding, with the matrix element encoding cells being additionally equipped with TO PSs in order to encode the $\left|X_i * w_{i,j}\right|$ product sign and ensure proper biasing of the nested MZIs[27]. Details of the experimental setup utilized for the chip characterization are provided in Supplementary section A.

### Silicon photonic Xbar Calibration

The effectiveness of a linear operator predominantly hinges on the programming precision of its nodes, that consequently translates into the accurate execution of the targeted linear matrix transformations. In our architecture, where EAMs comprise the basic constituent computational building blocks, our first step builds upon the assumption that all EAMs comprise identical devices. As such, we designed, fabricated and characterized a standalone EAM that shared the same design properties with the ones employed in the fabricated 4 × 4 Xbar of Fig. 1b. The experimentally measured performance metrics of the stand-alone EAM were used as a reference point for the induced IL and reverse bias versus attenuation relationship of the Xbar's constituent

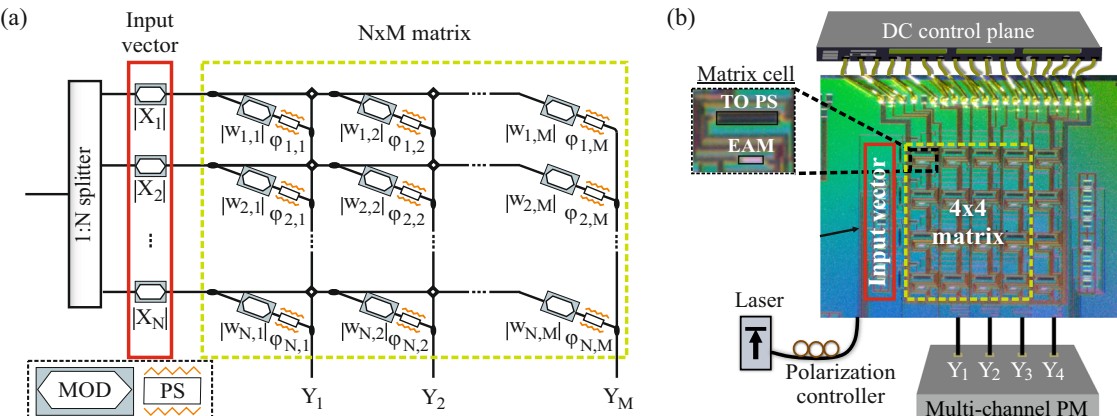

**Fig. 1 | Silicon Photonic Xbar layout and microscope photo. a** The N × M Crossbar architecture operating as a linear operator. **b** 4 × 4 Xbar chip, fabricated in silicon photonic platform. The yellow rectangle encompasses the NxM transformation matrix, while the input optical modes are represented by the red rectangle. Both input and matrix elements are realized employing SiGe EAMs.

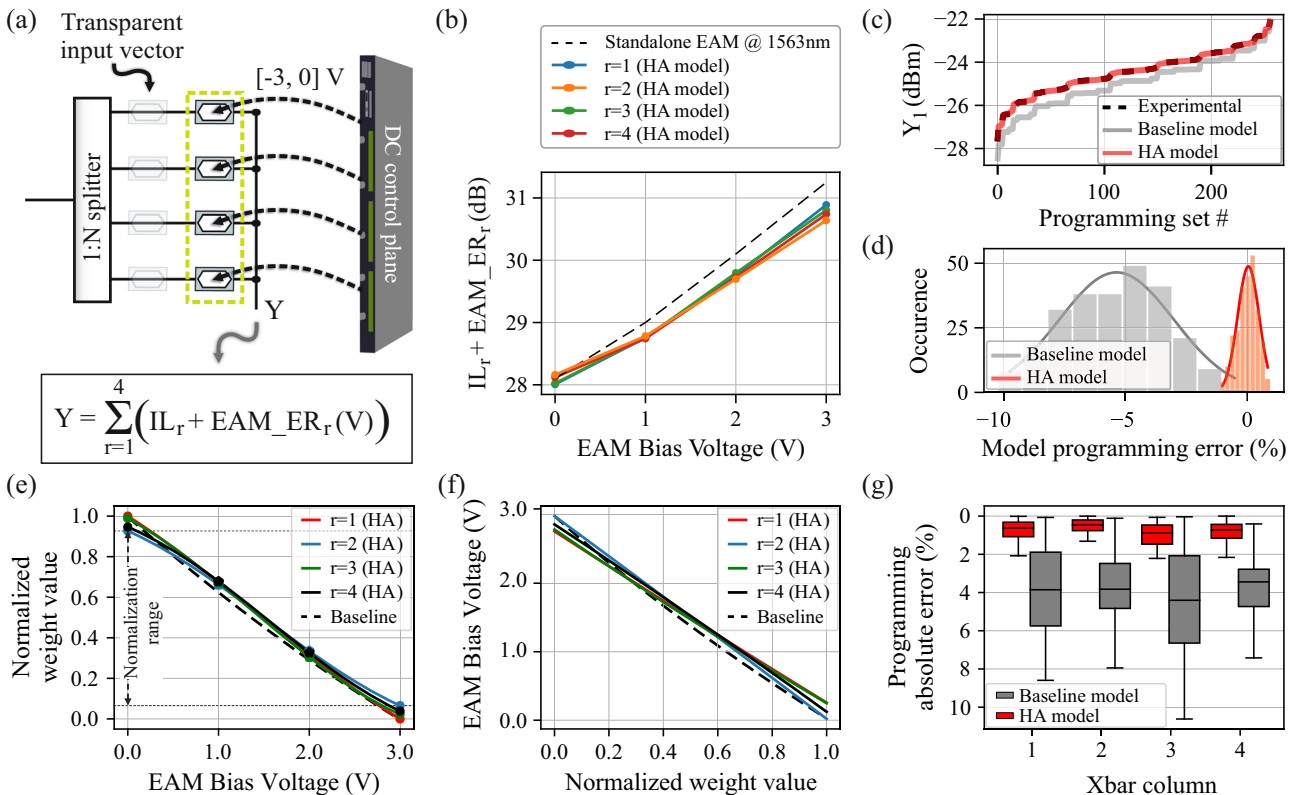

**Fig. 2 | Silicon photonic Xbar calibration. a** Testbed for the programming of Xbar's 1st column. The output Y comprises a linear combination of each Xbar row with IL and ER, corresponding to the excess loss introduced by each row and the attenuation induced by EAM, respectively. **b** Relation of Xbar rows' attenuation factor (photonic axes IL + EAMs extinction ratio) with their biasing voltage. **c** Experimental power levels obtained at Xbar's 1st column output for 256 different EAM biasing voltage sets and respective theoretically expected output power levels when using the baseline and HA programming models. **d** Baseline and HA model approximation error histograms. **e** Experimentally derived relation of row's weighting values versus EAMs bias voltage for the 1st column of the Xbar. **f** EAM bias voltage vs weight value "look-up" table. **g** Experimentally obtained absolute error when programming the Xbar by the baseline and HA model for all the columns. Error bars defined in [mean value, standard deviation]. Baseline model : Col1 [2.5,2.38], Col2 [4.91,1.75], Col3 [3.36,3.49], Col4 [3.78,1.6], HA model Col1 [0.64,0.47], Col2 [0.48,0.40], Col3 [0.87,0.6], Col4 [0.81,0.49].

EAM devices (see Supplementary section A). While this approach offers a baseline estimation of the Xbar's operational parameters, the unavoidable fabrication-induced performance variation between each constituent EAM, as well as the fabrication imperfections of the employed waveguide crossings and the horizontal and vertical light splitting stages, lead to imprecise mapping of the transformation matrix elements to the Xbar and, as a consequence, to imperfect fidelity transformations. Fidelity comprises a critical factor for linear optical circuits, representing a measure of closeness between the experimentally obtained values of a matrix and the intended ones.

The bijective weight-to-node mapping supported by the Xbar architecture allows us to compensate for the fabrication imperfections and to ensure a high-accuracy calibration procedure, taking advantage of the absence of any interdependence between the matrix nodes of different columns. A HA programming model was developed with its programming complexity analyzed in section B of the Supplementary material. Figure 2a illustrates the developed experimental testbed for the HA programming model when applied to the 1st column of our fabricated Xbar chip. With the input vector encoding EAMs being in their transparent state (V = 0), the direct current (DC) control plane was programmed to bias the node-EAMs of the 4 rows at discrete bias levels, within the range B = {0, − 1, − 2, − 3}V (see Supplementary section B). Following, the output Y was consecutively recorded via a power meter for all 256 (=4$^4$) possible linear transformations combination sets. When the nested MZIs of the 1st column get tuned to their fully-constructive interference state, then the column output equals $Y_1 = \sum_{r=1}^{4}(IL_r + EAM\_ER_r(V)), V \in B$, with $IL_r$ corresponding to the excess loss introduced by all column's components at row #r i.e., splitters, combiners, waveguide crossings, EAMs, and PSs, while $EAM\_ER_r(V)$ corresponds to the absorption-induced attenuation of the node-EAMs of row #r when biased with a voltage $V \in B$, which in turn, is correlated with the extinction ratio (ER). Given the unavoidable fabrication variation-induced discrepancy between the constituent $IL_r$ and $EAM\_ER_r(V)$ values and the baseline performance of the standalone EAM, we developed a linear regression algorithm (see "Methods" and Supplementary section B for more details) that exploits the experimentally obtained linear transformations to approximate the undetermined parameters.

In order to ease the understanding of the programming procedure, we indicatively illustrate in Fig. 2b the sum of $IL_r$ and $EAM\_ER_r(V)$ experimental values for the 4 constituent paths of the 1st Xbar column with r = [1,2,3,4]. These values were calculated via the HA model when a single node is driven in the range B = {0, − 1, − 2, − 3}V and the remaining 3 EAMs are driven at 0 V bias, revealing the divergence from the expected performance when using the standalone EAM metrics (baseline model). Figure 2c schematically captures the effectiveness of the developed HA programming model, by putting in juxtaposition the experimentally measured $Y_1$ output for the #256 linear transformation sets (black dashed line) with respect to the theoretically calculated $Y_1$ output when using the baseline programming model (gray curve) and the theoretically calculated $Y_1$ output when using the HA programming model (orange curve), revealing the excellent matching between the experimental curve and the curve calculated via the HA model. A quantitative representation of the model's accuracy in describing the

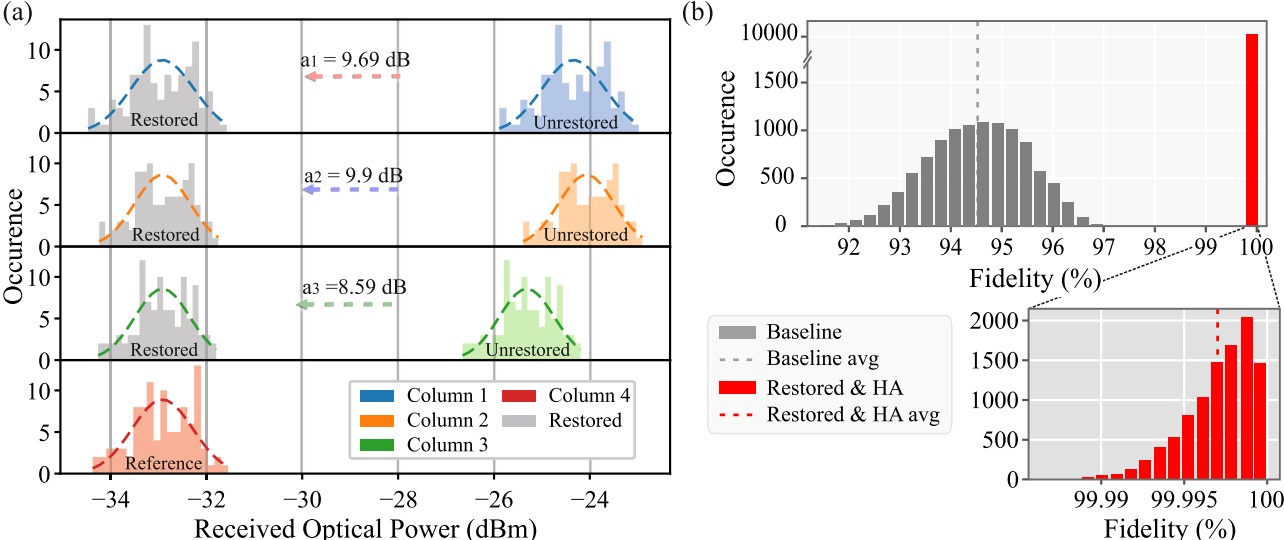

**Fig. 3 | Inter column fidelity restoration. a** Xbar fidelity restoration mechanism illustration for 0 dBm input power. The arrows, in the middle of the graph, represent the attenuation factors enforced at each column towards balancing the differential path loss among the Xbar columns. **b** Experimentally obtained fidelity in the execution of #10,000 arbitrary linear transformation matrices when the Xbar is programmed with the baseline model (gray) and the HA model and restoration procedure (red).

underlying hardware parameters is presented in Fig. 2d, where the error of the baseline and HA programming model outputs with respect to the experimentally obtained response is plotted with the gray and orange histograms, respectively. Fitting a Gaussian distribution into the acquired error values results in mean and standard deviation values that equal −5.45 and 2.04, and -2·10⁻³ and 0.41 for the baseline and HA models respectively, revealing the lack of a biased-error parameter in our HA-model and a significantly lower standard deviation. The same procedure was followed for the remaining 3 Xbar columns and 12 constituent EAM-based nodes, revealing a small deviation of <2% in the derived mean and standard deviation values.

Having estimated the individual parameters of the constituent EAM-based nodes with high-precision, we proceed by developing an algorithm that is capable of compensating for their performance divergence and providing the transformation matrix encoding map for arbitrary values. The first step includes the correlation of the required EAM bias voltage to each row's weighing factor by normalizing the $\left[\mathrm{IL}_r + \mathrm{EAM\_ER}_r(V)\right]$ values to [0, 1], as shown in Fig. 2e, where the same coloring with Fig. 2b was employed to discriminate between each row's respective curve. A 2nd order polynomial fit was applied to express the transfer function of each EAM, with high precision across the whole B range [0,3]. Thereafter, towards alleviating the inter-row weighting divergence, all 4 curves were bounded to the lowest and highest weight value of the 4 Xbar rows, highlighted in Fig. 2e, when the respective node-EAMs are biased at 0 and −3 Vs, respectively, and normalized again to the [0, 1] range. Finally, the targeted look-up table can be formulated by inverting the fitted transfer functions, as depicted in Fig. 2f. A set of #100 arbitrary transformation vectors were then generated and encoded at each Xbar column individually towards evaluating the accuracy of the developed HA programming model and the weighting calibration process. Figure 2g indicates the achieved absolute error values, normalized in percentage values through $E = \frac{P_{\mathrm{expected}} - P_{\mathrm{experimental}}}{P_{\mathrm{expected}}} * 100$, with $P_{\mathrm{expected}}$ corresponding to the expected output optical power for the given transformation and $P_{\mathrm{experimental}}$ to the measured optical power at the Xbar output, for a set of 100 arbitrary matrix transformations for all 4 Xbar columns, with the red and gray bars corresponding to the HA and baseline programming model, respectively. As can be observed, the HA model achieved an absolute

average error reduction of 3.2% compared to the baseline programming. Fitting a Gaussian distribution to the calculated error values, using their raw i.e., not-absolute values, concludes to mean and standard deviation values, expressed in normalized percentage values, of $\mu = [-0.39, -0.17, 0.04, 0.19]\%$ and $\sigma = [0.69, 0.6, 1.06, 0.92]\%$ for all 4 Xbar columns, respectively. In order to benchmark the achieved programming error versus the measurement uncertainty originating from the experimental setup, we performed a 1-hour long stability measurement using the experimental chain of Laser-Input GC-Chip Transmission-GC-PM, while applying 0 V to the constituent EAM and TO-PS. The measurements across the 4 column outputs revealed standard deviations in the range of [0.014, 0.028] dB that correspond to $\sigma_{\mathrm{equip}} = [0.3-0.6]\%$, while the noise limit of our experiment is dictated by the deployed measurement equipment. Specifically, we employed a commercially available 4-channel optical power meter (Keysight N7745A) with its specifications summarized in ref. 28 and corresponding to a typical noise value of σ (noise_limit) = -0.005%., significantly lower than the measured error. Therefore, the majority of the programming error originates from the environmental and mechanical perturbations during the PIC measurement procedure[29,30]. By implementing effective packaging strategies, it is possible to mitigate the impact of temperature variations and mechanical instability, and as such operate our Xbar prototype closer to its theoretical noise limits.

## Crossbar fidelity restoration & experimental results

### Inter-column fidelity restoration

Following the intra-column calibration procedure, we proceed by quantifying the fabrication error-induced inter-column differential loss and showcasing the functionality of the fidelity restoration mechanism of the Xbar photonic linear processor. Figure 3a illustrates the distribution of the optical powers emerging at all 4 Xbar column outputs, when 100 arbitrary matrices are experimentally enforced on an all-one input vector. Fitting each column's output power histogram to a gaussian distribution and comparing the resulting mean values allows for the quantification of the inter-column differential loss. As can be observed, the 4 Xbar columns exhibited an average optical loss

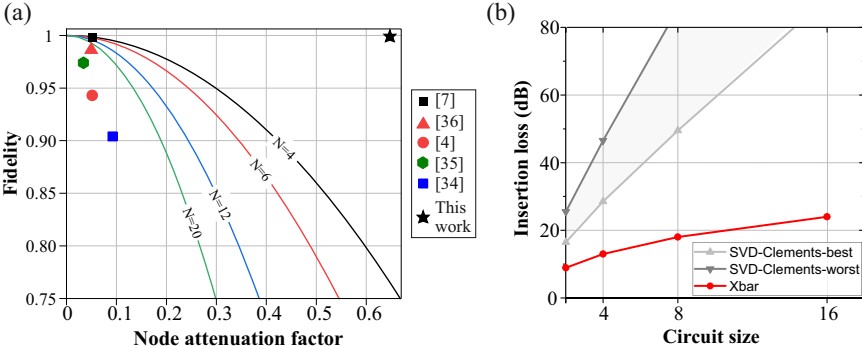

**Fig. 4 | Benchmarking versus state-of-the-art. a** Theoretically and experimentally obtained fidelity values of unitary-based architectures juxtaposed to the experimentally obtained fidelity performance of this work. **b** Theoretically expected IL values calculated using the proposed N × N Xbar (red) and SVD-Clements (gray) architectures for a node loss of 4.5 dB.

of 24.2 dB, 24.01 dB, 25.32 dB and 32.9 dB, respectively, with the maximum differential loss of 8.9 dB observed between columns #2 and #4 and owing mainly to the damaged I/O coupling in the 4th column. The average differential loss when only columns #1-#3 are considered ranged between [0.25 dB–1.32 dB], which is in line with the performance variation expected by fabrication variations at each constituent photonic building blocks, i.e., directional couplers, waveguide crossings, EAMs and MMIs. Despite the excess losses of column #4, the absence of interdependency between the elements of the Xbar output vector allows for the realization of an inter-column loss-balanced layout[14]. This can be readily achieved through selecting the gaussian fitted-mean value of the column with the lowest output power (column #4) and enforcing an attenuation factor $a_i$ at the output of every other $i$th column, with $\alpha_i$ equaling to the differential loss between column #4 and column #i. The addition of the external attenuation factors at the outputs of the three Xbar columns #1-#3 leads to a loss-balanced Xbar output vector and forms the fidelity restoration stage, as has been theoretically analyzed in ref. 14. A schematic representation of this procedure, when selecting column 4 as the reference column, is shown in Fig. 3a. The arrows, in the middle of the graph, correspond to the enforced attenuation factors $a_i$ and the gray-colored histograms to the column output power distributions after applying the fidelity restoration process.

### Experimental linear transformations fidelity of arbitrary matrices

The performance of the photonic Xbar as a linear operator was finally assessed by experimentally implementing a set of 10,000 [4 × 4] arbitrary transformation matrices, when considering an all-one input vector, and measuring the achieved fidelity. Fidelity has been defined as the Frobenious inner product of the targeted ($Y_{targ}$)) and the experimentally obtained ($Y_{exp}$)) output vectors, with the respective mathematical formula being: $F(Y_{exp}, Y_{targ}) = |tr(Y_{targ}^T * Y_{exp})/\sqrt{tr(Y_{targ}^T * Y_{targ}) * tr(Y_{exp}^T * Y_{exp})}|$. Figure 3b reveals the distribution of the achieved fidelities when the Xbar is programmed utilizing (i) a "baseline" matrix programming, that neither takes into account the fabrication variations of the constituent EAM-based nodes nor utilizes the fidelity restoration stage for differential loss balancing, and (ii) a fidelity restored and HA matrix programming that compensates for the fabrication imperfections. As can be observed our approach significantly increases the achieved fidelity from 94.35 ± 0.693% to an almost unity fidelity of 99.997 ± 0.002%, with the ±0.002% converging to the statistical error induced by the measurement procedure, highlighting in this way the substantial fidelity restoration credentials of our photonic Xbar architecture.

## Discussion

High-fidelity performance is a prerequisite for the majority of the targeted use-cases, becoming even more pronounced in specialized tasks, e.g., in quantum gates where >99% fidelity values are required[31], in safety critical NN workloads for automotive or unmanned aerial vehicle applications (UAVs)[32], and in high-precision optical beamformer networks for light detection and ranging (LiDAR) and microwave photonics[33]. In order to benchmark the experimentally demonstrated fidelity performance of our Xbar-based linear operator versus state-of-the-art approaches, Fig. 4a puts in juxtaposition: (i) The theoretical fidelity upper bound (see "Methods") versus node attenuation factor and for different radii between $N = 4$ and $N = 20$[4,7,34–36], for photonic processors that follow the prominent unitary-based architecture. Node attenuation factor is defined as $n_{Att} = 1 − 10^{ILnode/10}$, with ILnode denoting the IL per node.(ii) The experimentally reported fidelity of recent photonic linear processor prototypes, including our 4 × 4 Xbar demonstrator, with the scatter plot color corresponding to the prototype's scale and the respective theoretical curve. This analysis leads to three important conclusions (a) The achieved fidelity of an SVD-Clements optical processor heavily depends on both the scale and the IL of the constituent weighting nodes, effectively reaching a hard-limit based on the current SiPho platforms technology level. (b) The high-programming complexity of interdependent MZI-mesh architecture inevitably leads to high complexity calibration algorithms and subsequently to a discrepancy between the maximum theoretical limit and the achieved experimental fidelity, (c) The unique fidelity credentials of the proposed Xbar architecture are validated through our experimental demonstration, highlighting that our approach leads to easily calibrated and fidelity restorable layouts that are capable of breaking the scale plus IL/node tradeoff and reaching record high fidelities. This unique property can also unlock the employment of the Xbar architecture in applications where the operational speed is critical, i.e., data center (DC) cybersecurity[37], training of deep learning (DL) models[38,39], inference with tiled matrix multiplication etc[24], since it can tolerate the high-loss that is typically associated with high-speed optical modulators, without sacrificing fidelity performance. This architectural advantage extends also to the total circuit IL, as highlighted in Fig. 4b that illustrates the total IL values of both the Xbar and the SVD-Clements layouts for circuits sizes up to 64 and for a relatively high node IL value of ILnode = 4.5 dB, as has been the case in the fabricated 4 × 4 EAM-based Xbar chip. As can be observed, the linear scaling of the Xbar IL against the node IL[14] allows for practical implementations of up to 64 × 64 radii, considering a total IL limit of less than 40 dB. On the contrary, the use of node cascades in the case of the SVD-Clements layout leads rapidly to prohibitively high IL values for matrix sizes larger than 4 × 4.

Demarcating from coherent to incoherent architectures, ring-based layouts[21–23] have shown their capabilities to attain high fidelity values while retaining low programming complexity. Specifically, in ref. 23 the authors perform convolution operations in the synthetic frequency dimension harnessing the dynamics of light to implement highly compact devices. With regard to the calibration procedure, they utilize a linear regression model, i.e., the least-squares method, to ascertain the physical properties of their nodes, as has been the case of our Xbar architecture. However, in the case of Xbar, as the dimension of the targeted matrix increases, the calibration complexity similarly grows, scaling with $O(k \cdot N)^3$. In contrast, in the ring-based structure[23], the calibration complexity remains independent of the size of the convolution kernel. Yet, when employing ring-based layouts, additional challenges associated with temperature variation may arise, that should be taken into account towards maintaining high computation accuracies."

In summary, we presented a SiGe EAM-based Xbar architecture that can execute perfect universal linear optical operations, while also enabling ultra-high speed computations. By developing a generic, HA programming procedure, we experimentally performed 10,000 arbitrary linear transformations on a $4 \times 4$ SiPho chip with a record-high fidelity of $99.997\% \pm 0.002$ that converges to the statistical error induced by our measurement equipment. This performance allows for accurate experimental realizations of linear transformations, overcoming the inherent fidelity limits of $2 \times 2$ MZI-based interferometric unitary and universal layouts. Its experimental IL-fidelity performance reveals its unique credentials to sustain perfect linear operations even when high-loss optical nodes are employed, paving new inroads that expand along high-speed reconfigurable optical matrix realizations, while still retaining wavelength as an extra degree of freedom[35–41]. This can lead to new application segments, by operating either as an application-specific processor, following the graphics processing units (GPUs) paradigm, or as a universal linear processor by selecting the technological building blocks and its operational clock frequencies accordingly.

## Methods

### Experimental setup and SiPho processor characteristics

The $4 \times 4$ Xbar was fabricated in IMEC's ISSIPP50G SiPho platform using established building blocks from the supplied process design kit (PDK). The experimental setup is illustrated in Supplementary section A. A laser source was utilized to generate a continuous wave beam that was injected into the Xbar chip input port through a fiber array, while a polarization controller was employed to align the input polarization to the TE-optimized grating couplers, exhibiting an IL of -2.6 dB at the Xbar operating wavelength of 1563 nm. The residual ports of the fiber array were utilized to fan-out the 4 Xbar outputs that were converted to digital values via a multi-channel power meter (PM). The 50 μm-long 50 GHz SiGe EAMs, that were deployed as intensity modulators for the encoding of both the input vector and the transformation matrix elements, introduced an IL of 4.5 dB also at the 1563 nm operating wavelength. Silicon-based TO PSs were utilized for the sign imprinting at every matrix node and for biasing Xbar's nested MZIs. The Xbar chip, that occupies a total area of ~5.2 mm$^2$, was placed on a printed circuit board (PCB) that simplified the electrical interface between the electrically-controllable active components and the experimental testbed. More specifically, the DC pads of the SiPho chip were wirebonded in respective pads of the PCB and were routed to a break-out board through a ribbon cable. A commercial laser source[42], a polarization controller and an 8-channel power meter[43] facilitated the experimental testing, while a custom-made multi-channel digital-to-analog converter (DAC) was utilized to fine tune the TO PSs and encode the targeted transformation matrix elements onto the Xbar nodes according to a look-up-table that correlates the matrix values to the EAM driving voltages. Finally, it is worth noting that the optical path traversing the 4th column of the Xbar prototype introduced

significant higher losses than expected, a result attributed to a damaged I/O coupler.

### Xbar programming using linear algebra

The HA programming model for each Xbar column was trained using #256 experimentally obtained linear transformations, corresponding to the 256 possible configurations when biasing the four constituent EAMs of each column in the range B = [0,1,2,3]. This led to an overdetermined system of linear algebraic equations whose independent variables were the summation of the IL of each Xbar row $IL_r$ with the attenuation of each EAM $EAM\_ER_r(V)$ when biased with a voltage $V \in B$. In order to solve this system, we enforced the least squares regression model[44,45]. The least squares method is used to solve a linear system that has more equations than unknowns, by finding a solution that minimizes the sum of the squared errors between the expected and the experimentally obtained values, in contrast to simple linear algebra solvers that are designed to solve linear systems with a unique solution. More details on the complexity and the training requirements of the deployed model can be found in the Supplementary material section B.

### Fidelity analysis of SoTA architecture

In order to quantify the fidelity degradation introduced by the differential optical path losses of the Clements architecture[13], we calculated the achievable fidelity using a Monte-Carlo simulation analysis. Specifically, for different sets of circuit size N and node IL $IL_{node}$, with $N \in [4, 20]$ and $IL_{node} \in [0, 4.5]$ dB, we generated 500 random unitary matrices U and following the decomposition method described in ref. 13, we concluded to its expected hardware implementation $U_{exp}$. Thereafter, we calculated the fidelity of the latter for each set of N and $IL_{node}$ and fit the data into 2nd order polynomials to produce the curves shown in Fig. 4b.

## Data availability

The data that support the plots within this paper and other findings of this study are available from the corresponding authors upon request.

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

## Acknowledgements

We would like to thank Dr. Nikos Bamiedakis for fruitful discussions. The work was in part funded by the EU-project PlasmoniAC (871391) M.M.P., EU-project SiPHO-G (779664) M.M.P. and EU-project Gatepost (101120938) M.M.P.

## Author contributions

M.M.P., G.G., A.T. and N.P. conceived the experiment. M.M.P. and G.G deployed the experimental setup, performed the experiment and processed the experimental results. G.G., A.T. and N.P. performed the simulation analyses. D.L. and N.P. conceived the circuit architecture, while D.L., N.P and M.M.P. contributed to the mask design. All authors discussed the results and wrote the manuscript.

## Competing interests

The authors declare no competing interests.
