## [Peer Review File · Nature Communications]

Perfect Linear Optics using Silicon PhotonicsREVIEWER COMMENTS

Reviewer #1 (Remarks to the Author):

The authors fabricated a coherent Si crossbar which could support on-chip fidelity Restoration. They experimentally demonstrated 10000 arbitrary linear transformations with a high fidelity of 99.997%. The work may be interesting, but the current version could not be accepted. The manuscript may be more suitable for the journal of Communications Physics.

Other suggestions:

1 about the writing. The readability of the manuscript is too poor.

(1) Since the word "silicon photonics" has definite meanings, the word "SiPho crossbar (Xbar)" is very odd and difficult to understand for a broad community.

(2) The meaning of the word "fidelity" should be defined in the introduction part.

(3) Some figure caption is too short and unclear expression. For example, "Fig. 1: (a) N×M Crossbar architecture. (b) 4×4 SiPho Xbar chip."

(4) There are too many abbreviations without definition.

2 about the novelty.

Ref. 14 has reported the SiPho Xbar in detail, which is authors' previous work. In the Abstract of this manuscript, the authors claimed "we present for the first time a novel coherent SiPho crossbar (Xbar).....". do you think this claim is suitable?

The novelty of this work is experimental demonstration of arbitrary linear transformations, i.e. applications of Ref.14. Even authors claimed "for the first time, to the best of our knowledge", I do not think the manuscript has enough breakthrough.

Reviewer #2 (Remarks to the Author):

In “Perfect Linear Optics using Silicon Photonics”, the authors applied coherent silicon crossbar techniques to overcome fabrication imperfections. Authors experimentally show the accurate mapping of a linear operator matrix into silicon photonic mesh. Examples using Xbar for various linear operations based on an on-chip crossbar structure are used to exemplify and elaborate the results. A hardware-aware (HA) model of insertion loss and unity transformation has been proposed in the optical domain to explain the results. Based on the models, the matrix mapping accuracy has reached almost unity, only constrained by the measurement error.

While the technical results look correct, the overall presentation is a bit misleading. The applicability of these results in real-world settings is also questionable. Some of the considerations below may eliminate the novelty claims as presented by the authors. In view of these, I have several concerns listed below and suggest major revisions to align with the novelty criteria of Nature Communications. Alternatively, I suggest the authors consider a more specialized journal.

1. The authors have used the simplified hardware-aware model to describe the silicon photonic chip. However, this simplification cannot be regarded as $O(N)$, as this is only the lowest order simplification, ignoring many important physical processes which may play an important role of computation accuracy. Also, the authors only explain the case when this model is applied to the first column of the Xbar mesh, namely Y1. This makes the validity of applying to the entire chip unclear. It will be clearer if a formula is shown to describe the procedure when extending from Fig. 2 (e) to Fig. 2 (g). Can the authors provide a more detailed justification and thorough discussion of the HA model?

2. It is well known that the MZI meshes, e.g. Clements structure shown in the paper as juxtaposition, are hard to control with a heat phase shifter. Therefore MZI meshes exhibit inherent architectural deficiencies that mainly originate from the accumulating nature of the fabrication imperfections across their cascaded nodes. However, the weight and phase control shown in the paper are very close to the programming mechanism in the ring resonator-based computation structures. Two examples: PhysRevApplied.18.034088 and sciadv.adi4956, among others. In these types of structures, the programming is also more robust to fabrication errors as 1. The voltage control can be very precise electronically, and 2. The ring can be fabricated in high accuracy with a very high quality factor. Besides, this ring-based structure can easily incorporate hundreds of frequency comb lines, instead of 4x4 inputs shown in this paper. A discussion and comparison of the crossbar and ring-based structure in terms of computation robustness are necessary in the introduction and discussion sections of the paper.

3. The most important characteristic of an accurate mapping of a linear matrix is the robustness against fabrication imperfection. This is known as error correction for fabricated components of an optical neural network. The HA model dictates the insertion loss limit and the restoration programming from the voltage lookup table. While the authors acknowledged that the fidelity description is only characterized by a particular crossbar structure shown in the paper, it is questionable how generalizable it is for a broad range of samples. Could the authors show a large number of samples and the variance of the result compared in a given pool of structures?

4. Related to the previous point, the measurement is the reason that the fidelity cannot go higher. Since there are always noise in measurement, shot noise, environment noise, thermal noise, quantum noise etc. Could the authors give a model for the noise limit? What is the dynamic range of the proposed measurement scheme? Could the range with large slopes in Fig. 2 extend over a large range of voltage?

Reviewer #1

The authors fabricated a coherent Si crossbar which could support on-chip fidelity Restoration. They experimentally demonstrated 10000 arbitrary linear transformations with a high fidelity of 99.997%. The work may be interesting, but the current version could not be accepted. The manuscript may be more suitable for the journal of Communications Physics.

Comment #1:

About the writing. The readability of the manuscript is too poor.

(1) Since the word “silicon photonics” has definite meanings, the word “SiPho crossbar (Xbar)” is very odd and difficult to understand for a broad community.

(2) The meaning of the word “fidelity” should be defined in the introduction part.

(3) Some figure caption is too short and unclear expression. For example, “Fig. 1: (a) $N \times M$ Crossbar architecture. (b) 4×4 SiPho Xbar chip.”

(4) There are too many abbreviations without definition.

P

Reply-

We would like to thank the Reviewer for his/her remarks and recommendations. The revised manuscript incorporates all the related changes including the fidelity definition in the intro section.

Changes in the text:

Introduction section line 42 the following text has been modified :

*” This has a significant impact to their size scaling and overall performance credentials¹⁴ i.e., fidelity, **defined as the Frobenious inner product of the targeted and the experimentally obtained output vectors**, programming complexity, processing bandwidth, energy- and footprint efficiency”*

Comment #2:

Ref. 14 has reported the SiPho Xbar in detail, which is authors’ previous work. In the Abstract of this manuscript, the authors claimed “we present for the first time a novel coherent SiPho crossbar (Xbar).....”. do you think this claim is suitable?

The novelty of this work is experimental demonstration of arbitrary linear transformations, i.e. applications of Ref.14. Even authors claimed “for the first time, to the best of our knowledge”, I do not think the manuscript has enough breakthrough.

Reply:

We would like to thank the Reviewer for his/her comments and for giving us the opportunity to better clarify the novelty of this manuscript.

More precisely, Ref. 14 of the original manuscript introduced for the first time the Xbar architecture as a layout and presented its theoretical foundations and theoretical performance analysis. However, this work has been **solely restricted to the circuit architecture without reporting neither any integrated Xbar circuit prototype nor any experimental evidence** of its

theoretically obtained performance. As such, our current work is **not only** the first experimental demonstration of arbitrary linear transformations, as the reviewer indicates, **but is also the first that** a) implements the Xbar architecture as a silicon-integrated chip, b) validates experimentally that it can guarantee an almost perfect fidelity performance. Transforming a theoretical work into an experimental demonstration comprises a highly challenging and innovative work on its own, since the theoretically obtained performance assumes almost ideal photonic components and gets confronted with several challenges until translating into an experimentally obtained performance, where non-ideal fabricated photonic devices are employed. This consensus is widely accepted in the research community, with an indicative example from the field of linear optics being the transfer of the pioneering architectural work of Clements et al. [1] into an experimental demonstration [2], where despite the authors' efforts to address the challenges associated with fabrication imperfections, the experimentally achieved fidelity exhibited a degradation of >2.5% compared to the theoretically predicted performance due to the deployment of imperfect components. To this end, both the theory as well as the experimental chip-scale deployment of the Clements unitary layout were appreciated as highly innovative developments and both are currently considered as pioneering and highly-cited articles in the field.

In the case of the Xbar architecture, the experimental demonstration requires, among others, time synchronization of the optical modes, accurate measurement and control of the input and output signals, and systematic calibration and compensation of the node parameters. Left unattended, these challenges will inevitably degrade the achieved performance and widen the gap between the theoretically predicted and experimentally obtained fidelity of the Xbar circuitry. In this context, we believe that our manuscript does not only address the challenges associated with transferring an optical architecture into a Silicon Photonic platform but at the same time offers significant breakthroughs in the field of linear optics. More specifically:

- i. We present and experimentally verify a novel hardware-aware (HA) model for calibrating our Xbar architecture. Our approach leverages the single-column dependence of each Xbar's output to achieve low programming error and high-converge to the physical properties of each constituent Xbar node.
- ii. We experimentally demonstrate for the first time almost unity on chip fidelity restoration, reaching the highest achieved so far experimental fidelity performance of 99.997%, while also experimentally validating the fidelity restoration capabilities of our Xbar architecture.
- iii. Our proposed architecture comprises the only linear operator in the literature that achieves high-fidelity values without necessitating the deployment of ultra-low loss nodes, opening up new horizons in the domain of universal linear optics and potentially engendering new applications and significant breakthroughs.

We hope that the above analysis has addressed the reviewer concerns and has clarified the novelty of our work and our claims. Apart from the changes introduced in the revised manuscript in order to address the Reviewer's concerns and improve our manuscript, we would also like to refer the Reviewer to the extended analysis of our HA-model provided in the revised Supplementary section, that we believe further strengthens our manuscript's impact.

Changes made in the text:

Abstract section line 17 the following text has been modified :

“In this work, we present for the first time the experimental deployment of a 4×4 coherent crossbar (Xbar) as a silicon chip and validate experimentally its theoretically predicted fidelity restoration credentials”

Introduction section line 79 the following text has been modified :

“The fabricated 4×4 silicon photonic Xbar employs 50GHz silicon germanium (SiGe)-based electro-absorption modulators (EAMs) for the encoding of both the 4-element input vector and the 4×4 transformation matrix, using in addition silicon-based thermo-optic (TO) phase shifters (PSs) for fine phase control... “A novel hardware calibration model is proposed and employed for accurately assessing the performance of each individual EAM-based Xbar matrix cell, substantially decreasing the programming error compared to the baseline performance”

Reviewer #2

In “Perfect Linear Optics using Silicon Photonics”, the authors applied coherent silicon crossbar techniques to overcome fabrication imperfections. Authors experimentally show the accurate mapping of a linear operator matrix into silicon photonic mesh. Examples using Xbar for various linear operations based on an on-chip crossbar structure are used to exemplify and elaborate the results. A hardware-aware (HA) model of insertion loss and unity transformation has been proposed in the optical domain to explain the results. Based on the models, the matrix mapping accuracy has reached almost unity, only constrained by the measurement error. While the technical results look correct, the overall presentation is a bit misleading. The applicability of these results in real-world settings is also questionable. Some of the considerations below may eliminate the novelty claims as presented by the authors. In view of these, I have several concerns listed below and suggest major revisions to align with the novelty criteria of Nature Communications. Alternatively, I suggest the authors consider a more specialized journal

Comment #1:

The authors have used the simplified hardware-aware model to describe the silicon photonic chip. However, this simplification cannot be regarded as $O(N)$, as this is only the lowest order simplification, ignoring many important physical processes which may play an important role of computation accuracy. Also, the authors only explain the case when this model is applied to the first column of the Xbar mesh, namely Y1. This makes the validity of applying to the entire chip unclear. It will be clearer if a formula is shown to describe the procedure when extending from Fig. 2 (e) to Fig. 2 (g). Can the authors provide a more detailed justification and thorough discussion of the HA model?

Reply:

We would like to thank the Reviewer for his insightful comments and apologize for any misunderstanding considering the computational complexity of our calibration model.

In order to clarify our approach, we begin by describing the differences between the programming and calibration computing complexity of a photonic matrix and report the respective metrics. The programming complexity is associated with the procedure required to imprint an arbitrary transformation matrix onto the photonic hardware. In the prominent SVD-based approach, transferring a $N \times N$ transformation matrix into the photonic accelerator necessitates the calculation of the constituent matrices of the SVD decomposition scheme, a mathematical operation that typically has a programming complexity of $O(N^3)$ [3]-[4]. In the case of our proposed photonic Xbar architecture, its bijective nature implies that the programming complexity is reduced to a memory look-up operation, as each weight node has a unique relationship with the targeted weight matrix coefficients. This operation for a single node has a computational complexity of $O(1)$, a complexity that can be maintained for a $N \times N$ matrix when assuming a parallel computational system or reach $O(N)$ when assuming column parallel memory look-up as presented in the original manuscript.

On the other hand, the calibration complexity of a photonic matrix is related to the algorithm that approximates and assimilates in the programming procedure, the inevitable fabrication-induced performance variations of the constituent matrix nodes. The related complexity for SVD-based approaches has been extensively investigated, with an interesting comparative study presented in Appendix B of [5]. In the case of our HA-model, the calibration complexity is mainly

dictated by the computational complexity of the least-squares method applied to calculate the vector \vec{C} , that expresses the transmission coefficients of each constituent node. Generalizing the procedure described in part B of the Supplementary material for an $N \times N$ Xbar array and k possible values for the EAM voltage space, such as $B = \{0_1, \dots, -3_k\} V$, we can conclude to the following linear system:

$$\vec{Y} = \mathbf{T} * \vec{C} \quad (1)$$

with \vec{Y} representing the output vector with a dimensionality of $[k^N \times 1]$ and k^N related to the possible combinations of voltages applied in each Xbar column constituent EAM nodes, reaching up to $-3V$ in order to maintain a safety margin with their breakdown voltage, \mathbf{T} corresponding to the binary-based expression of the node contribution with a dimensionality of $[k^N \times k \cdot N]$ and the unknown vector \vec{C} expressing the k coefficients for each constituent node with dimension of $[k \cdot N \times 1]$. Expressing equation 1 including the related dimensions leads to:

$$\vec{Y}[k^N \times 1] = \mathbf{T}[k^N \times k \cdot N] * \vec{C}[k \cdot N \times 1] \quad (2)$$

with the computational complexity of solving equation (2) using the least-squares method mainly dictated by the dimensionality of the \mathbf{T} matrix and typically approximated with a complexity of $O(N^3)$ for an $M \times N$ matrix [6]. As such, the generalized form of the computation complexity of the HA-model is $O(k \cdot N)^3$. From this expression we can deduct that the main contributors to the computational complexity are: (i) The value k representing the number of values describing the EAM-based node transfer function (ii) The row dimension of the matrix \mathbf{T} , dictated by both the k value and also the required number of equations for approximating \vec{C} with high precision.

Considering the required number of k values to accurately approximate the transfer function of the EAM-based node, we plot in Fig. 1 (a) the normalized transfer function of a stand-alone 50 μm long EAM for three different wavelengths, located in the periphery of its maximum FOM operating point. The experimentally derived transfer function comprised 12 points equally spaced in the range $R = [0,3] V$, with R representing the applied reverse bias voltage. Figure 1 (b) illustrates the mean absolute error achieved when fitting an increasing number of equally space data points using a 2nd order polynomial and comparing with the experimentally derived performance. As we can observe, even with $k \leq 4$ the related improvement in precision is less than 0.3% for all 3 wavelengths, showcasing that even 4 data points are sufficient to accurately model an EAM-based node. Finally, as analyzed in Section B of the supplementary material, due to the over parametrized

Fig. 1: (a) Normalized 50 μm EAM transfer function for three different wavelengths (b) Absolute mean error versus number of points used for fitting a 2nd order polynomial to the experimental data.

nature of our approach, we can also reduce the row dimension of the T matrix, trading precision with required calibration time and computational complexity.

Considering the transition from Fig. 2 (e) to Fig. 2 (g) we have included in the supplementary material a programming diagram of the related experimental procedure, while also extending the presented results for all the columns of the 4×4 Xbar.

Changes made in the text:

Section B line 122 the following text has been modified.

“A ~~low-complexity $O(N)$~~ hardware-aware (HA) programming model was developed with its programming complexity analyzed in section B of the Supplementary material. Figure 2 (a) illustrates the developed experimental testbed for the HA programming model when applied to the 1st column of our fabricated ~~SiPho~~ Xbar”

Based on the Reviewer’s comment we have updated the supplementary material with the following sections

A. Hardware aware programming model : Algorithm and computational complexity

[a]. Algorithm description

Figure S3 illustrates a generalized form of the algorithmic procedure followed during the application of the HA-model, with the brackets corresponding to the algorithmic description:

HA-model algorithm	
r :	Crossbar row
c :	Crossbar column
φ :	Phase shift
w :	Weight
EAM_rb :	EAM reverse bias voltage (V)
1:	For r, c in [1, 4]:
	Tune $\varphi(r, c)$.
2:	For r in [1, 4]:
	For c in [1, 4]:
	For $EAM_rb(r, c)$ in [0, 3]:
5a:	Measure $Y = \Sigma(c, EAM_rb(r, c))$
5b:	Normalize to Σ to [0, 1]
6:	Solve $Y = T * C$ to get w coefficients
7:	For c in [1, 4]:
	For r in [1, 4]:
9a:	Limit $w(r, c, EAM_rb)$ to [$\min(w([1, 4], c, 0), \max(w([1, 4], c, 3))$]
9b:	Normalize to [0, 1]
10:	Find $EAM_rb(r, c, w)$

Fig. S3: HA-model algorithmic description

1. [1]. The PSs of each Crossbar column are fine tuned so that nested MZIs interfere constructively.

2. [2-5b] Pilot linear transformation measurements are performed, by permutating the voltages of each column’s constituent EAM-based nodes. The end results are normalized to [0,1].

3. [6] The least square method is employed to calculate the weight contribution of each row node as a function of the applied reverse voltage.

4. [9a-9b] Limit each column’s constituent EAM operating voltages to [minimum achievable transmission of all row contributions at -3V, maximum achievable transmission of all row contributions at 0V] and normalize the vectors to [0,1].

5. [10] (i) Find the row contribution vs EAM bias voltage fit and calculate the inverse transfer functions to transform voltage values to weighing factors and find each EAM’s voltage range (ii) Normalize transmission on new EAMs bias voltage ranges and “fine-fit”

the rows' contribution vs EAM bias voltage relation. (iii) Normalize the rows contribution values to [0, 1] and calculate the inverse transfer function. The latter corresponds to the required look-up table between the weight values and the EAM-voltages.

The related results for all four Xbar's columns, after the application of the least-square method and the final look up tables are illustrated in Fig. S4 (a) and (b) respectively.

Fig. S4: (a) Experimentally derived relation of row's weighting values versus EAMs bias voltage for all 4 Xbar columns. (f) EAM bias voltage vs weight value "look-up" table for all 4 Xbar columns

[b]. Computational complexity

In order to clarify our approach, we begin by describing the differences between the programming and calibration computing complexity of a photonic matrix and report the respective metrics. The programming complexity is associated with the procedure required to imprint an arbitrary transformation matrix into the photonic hardware. In the prominent SVD-based approach, transferring a $N \times N$ transformation matrix into the photonic accelerator, necessitates the calculation of the constituent matrices of the SVD decomposition scheme, a mathematical operation typically having a programming complexity of $O(N^3)$ [3][4]. In the case of our proposed photonic Xbar architecture, its bijective nature implies that the programming complexity is reduced to a memory look-up operation, as each weight node has a unique relationship with the targeted weight matrix coefficients. This operation for a single node has a computational complexity of $O(1)$ [5], a complexity that can be maintained for a $N \times N$ matrix when assuming a scaled memory look-up system.

On the other hand, the calibration complexity of a photonic matrix is related to the algorithm that approximates and assimilates in the programming procedure the inevitable fabrication-induced variations. The related complexity for SVD-based approaches has been extensively studied, with an interesting comparative study presented in Appendix B of [6]. In the case of our HA-model, the calibration complexity is mainly dictated by the computational complexity of the least-squares method applied to calculate the vector \vec{C} , that expresses the transmission coefficients of each constituent node. Generalizing the procedure described in part B (a) of the Supplementary material for an $N \times N$ Xbar array and k possible values for the EAM voltage space, such as $B = \{0_1, \dots, 3_k\} V$, we can conclude to the following linear system:

$$\vec{Y} = \mathbf{T} * \vec{C} \quad (1)$$

,with \vec{Y} representing the output vector with a dimensionality of $[k^N \times 1]$ and k^N related to the possible combinations of voltages applied in the each Xbar column constituent EAM nodes, reaching up to -3V reverse bias voltages for safety reasons, T corresponding to the binary-based expression of the node contribution with a dimensionality of $[k^N \times k \cdot N]$ and the unknown vector \vec{C} expressing the k coefficients for each constituent node with dimension of $[k^N \times 1]$. Expressing equation 1 including the related dimensions leads to:

$$\vec{Y}[k^N \times 1] = T[k^N \times k \cdot N] * \vec{C}[k \cdot N \times 1] \quad (2)$$

with the computational complexity of solving equation (2) using the least-square method mainly dictated by the dimensionality of the T matrix and typically approximated with a complexity of $O(N^3)$ for an $M \times N$ matrix [Σφάλμα! Το αρχείο προέλευσης της αναφοράς δεν βρέθηκε.]. As such, the generalized form of the computation complexity of the HA-model is $O(k \cdot N)^3$. From this expression we can deduct that the main contributors to the computational complexity are: (i) The value k represents the number of values describing the EAM-based node transfer function (ii) The row dimension of the matrix T , dictated by both the k value and also the required number of equations for approximating \vec{C} with high precision.

Considering the required number of k values to accurately approximate the transfer function of the EAM-based node, we plot in Fig. S5 (a) the normalized transfer function of a stand-alone 50um long EAM for three different wavelengths, located in the periphery of its maximum FOM operating point. The experimentally derived transfer function comprised 12 points equally spaced in the range $R = [0,3]$ V, with R representing the applied reverse bias voltage. Figure S5 (b) illustrates the mean absolute error achieved when fitting an increasing number of equally space data points using a 2nd order polynomial and comparing with the experimentally derived performance. As we can observe, even with $k \leq 4$ the related improvement in precision is less than 0.3% for all 3 wavelengths, showcasing that even 4 data points are sufficient to accurately model an EAM-based

Fig. S5: (a) Normalized 50-um EAM transfer function for three different wavelengths (b) Absolute mean error versus number of points used for fitting a 2nd order polynomial to the experimental data.

node.

Comment #2:

It is well known that the MZI meshes, e.g. Clements structure shown in the paper as juxtaposition, are hard to control with a heat phase shifter. Therefore MZI meshes exhibit inherent architectural deficiencies that mainly originate from the accumulating nature of the fabrication imperfections across their cascaded nodes. However, the weight and phase control shown in the paper are very close to the programming mechanism in the ring resonator-based computation structures. Two examples: PhysRevApplied.18.034088 and sciadv.adi4956, among others. In these types of

structures, the programming is also more robust to fabrication errors as 1. The voltage control can be very precise electronically, and 2. The ring can be fabricated in high accuracy with a very high quality factor. Besides, this ring-based structure can easily incorporate hundreds of frequency comb lines, instead of 4x4 inputs shown in this paper. A discussion and comparison of the crossbar and ring-based structure in terms of computation robustness are necessary in the introduction and discussion sections of the paper.

Reply:

The demonstrations in [7] describe theoretically and experimentally, respectively, a novel way of performing convolution operations in the synthetic frequency dimension leading to highly compact devices. Specifically, the authors utilize a ring resonator modulator to synthesize arbitrary convolution kernels and demonstrate the convolution computation between input frequency combs and synthesized kernels. The proof-of-concept demonstration [8] was experimentally validated using fiber-based components revealing an excellent agreement with respect to the theoretical predictions, thanks to the deployment of a least-squares-based calibration method that determines the system's loss rate and the optimal modulation waveform and in this way ensures computation robustness of the desired convolution kernels. Demarcating, however, from a bulky fiber-based demonstration to a fully integrated system in a single photonic integrated circuit, one should take into account some additional challenges that may affect the computation accuracy of the system i.e., a) the frequency comb line power may fluctuate over wavelength and time and lead to variation in the convolution output and b) temperature variations would necessitate precise control of the ring and frequency comb resonances.

To this end, it becomes clear that it is rather unfair to compare these two architectures on an equivalent basis, since i) the ring-based structure harnesses the synthetic frequency dimension and WDM capabilities of light, retaining a rather simple single-ring layout by shifting all the complexity to the unit required for highly accurate multi-channel lasing, ii) while the Xbar architecture exploits interference for performing linear operations relying always on a single-wavelength channel. This implies that our layout should be assessed within the framework of coherent architectures, as has been typically the case so far in all different coherent linear optical demonstrations [1],[9]-[10]. This has been also the primary rationale for utilizing the SVD-Clements architecture performance as a comparison baseline. It is worth noting that the Xbar architecture can support operations in both space and wavelength dimensions as well [Ref Angelina] to scale up the computational throughput.

Nevertheless, we agree with the Reviewer's statement that the ring-based structure can in principle incorporate hundreds of frequency comb lines, thereby enhancing the matrix scalability credentials. However, this enhancement comes at the cost of imposing limitations on operational speeds, as spurious interference effects may arise due to the necessity of keeping the bandwidth of the photodetector lower than the deployed comb spacing. Moreover, in a fully-integrated system, the challenge of accommodating a large free spectral range of ring modulators persists, presenting a significant constraint on the simultaneous coverage of modes by both modulators and photodetectors. Finally, the ring-based demonstration has been specifically designed to optimally perform neural network convolution operations, while the Xbar architecture can be adopted for a broad range of applications, including microwave photonics, quantum and optical computing.

With regard to the calibration procedure, as the Reviewer aptly highlighted, both the proposed Xbar architecture and the ring-based structure utilize a linear regression model (least-squares method) to ascertain the physical properties of their nodes. However, in the case of Xbar, as the

dimension of the targeted matrix increases, the calibration complexity similarly grows, scaling with $O(k \cdot N)^3$ (analyzed in Comment #1 of 2nd Reviewer). In contrast, in the ring-based structure, the calibration complexity remains independent of the size of the convolution kernel.

In order to provide this information in our manuscript, complying with the reviewer's comment, we include the following changes in our text:

Changes made in the text:

Introduction section, line 68 the following text has been added:

“On the contrary, in ring-based architectures²¹⁻²³, that harness the synthetic frequency dimension and/or wavelength division multiplexing (WDM) capabilities of light, programming comprises a straightforward procedure that can increase their computation robustness and fidelity performance. Yet, in these implementations the circuit complexity shifts to the unit required for generating multiple optical channels; this gets even more pronounced when targeting a fully-integrated system, where the need for multiple wavelength channels and as such resonant devices will inevitably impose significant scalability challenges, since the temperature variation would necessitate precise control of the ring resonances.”

Discussion section, the following text has been added:

“Demarcating from coherent to incoherent architectures, ring-based layouts²¹⁻²³ have shown their capabilities to attain high fidelity values while retaining low programming complexity. Specifically, in²³ the authors perform convolution operations in the synthetic frequency dimension harnessing the dynamics of light to implement highly compact devices. With regard to the calibration procedure, they utilize a linear regression model, i.e the least-squares method, to ascertain the physical properties of their nodes, as has been the case of our Xbar architecture. However, in the case of Xbar, as the dimension of the targeted matrix increases, the calibration complexity similarly grows, scaling with $O(k \cdot N)^3$. In contrast, in the ring-based structure²³, the calibration complexity remains independent of the size of the convolution kernel. Yet, when employing ring-based layouts, additional challenges associated with temperature variation may arise, that should be taken into account towards maintaining high computation accuracies.”

Comment #3:

The most important characteristic of an accurate mapping of a linear matrix is the robustness against fabrication imperfection. This is known as error correction for fabricated components of an optical neural network. The HA model dictates the insertion loss limit and the restoration programming from the voltage lookup table. While the authors acknowledged that the fidelity description is only characterized by a particular crossbar structure shown in the paper, it is questionable how generalizable it is for a broad range of samples. Could the authors show a large number of samples and the variance of the result compared in a given pool of structures?

Reply:

Based on the Reviewer comment we employed our HA programming model in two additional 4×4 Xbar chips. Figure 2 (a)-(c) puts in juxtaposition the experimentally obtained absolute error when programming the Xbar chips using the baseline and the HA model for all the columns. The reported absolute mean error and std values for all the 4 columns of the 3 different chips for the HA-model were measured to be $\mu=[0.74,0.81,1.1]$ and $\sigma=[0.48,0.53,0.8]\%$, showcasing a small divergence range and validating the capabilities of the HA model in minimizing the respective programming error.

Figure 2 (d) illustrates a histogram of the experimentally acquired fidelity values for 10,000 arbitrary matrix transformation for all 3 Xbar chips. The 3 different chips achieved matrix fidelities of $99.997\pm 0.002\%$, $99.995\pm 0.003\%$, $99.992\pm 0.005\%$ respectively, validating the potential of our HA-model and fidelity restoration mechanism in achieving almost unity fidelity values with small variance across the different samples. At this point we would also like to mention that the application of the HA-model across 3 different chips was not arbitrary, but corresponded

Fig. 2: (a) - (c) Experimentally obtained absolute error when programming the Xbar by the baseline and HA model for all columns and 3 different chips. (d) Experimentally obtained fidelity in the execution of #10,000 arbitrary linear transformation matrices for 3 different chips, after employing the HA-programming model and fidelity restoration

to the total packaged Silicon Photonics chips we had available. Concluding to more significant statistical values would necessitate wafer-scale validation, a topic that we believe could be the scope of a future study.

Changes made in the text:

The following sub-section has been added to the supplementary material :

[a]. Validation across multiple chips

In order to assess the effectiveness of our HA-programming model in a larger number of samples, two additional 4×4 Xbar chips were measured and calibrated. Figure S7 (a)-(c) puts in juxtaposition the experimentally obtained absolute error when programming the Xbars using the baseline and HA model for all the columns. The reported absolute mean error and std values for all the 4 columns of the 3 different chips for the HA-model were measured to be $\mu=[0.74,0.81,1.1]$ and $\sigma=[0.48,0.53,0.8]\%$, showcasing a small divergence range and validating the capabilities of the HA model in minimizing the respective programming error. Figure S7 (d) illustrates a histogram of the experimentally acquired fidelity values for 10.000 arbitrary matrix transformation for all 3 Xbar chips. The 3 different chips achieved matrix fidelities of $99.997\pm 0.002\%$,

Fig. S7: (a) - (c) Experimentally obtained absolute error when programming the Xbar by the baseline and HA model for all columns and 3 different chips. (d) Experimentally obtained fidelity in the execution of #10000 arbitrary linear transformation matrices for 3 different chips, after employing the HA-programming model and fidelity restoration.

$99.995\pm 0.003\%$, $99.992\pm 0.005\%$ respectively, validating the potential of our HA-model and fidelity restoration mechanism in achieving almost unity fidelity values with small variance across different samples.

Comment #4:

Related to the previous point, the measurement is the reason that the fidelity cannot go higher. Since there are always noise in measurement, shot noise, environment noise, thermal noise, quantum noise etc. Could the authors give a model for the noise limit? What is the dynamic range of the proposed measurement scheme? Could the range with large slopes in Fig. 2 extend over a large range of voltage?

Reply:

As with any processing system, calculating the noise limit of our accelerator, necessitates the definition of the major contributing noise sources that directly define the respective noise-limited operating regime. To this end, one of our previous studies [11] analyzed the major electronic and optical noise sources and the respective theoretically operating boundaries impacting a Xbar photonic accelerator, revealing the trade-offs between operational compute rate and precision of matrix transformations. In our current experimental setup, the noise limit is not only related to these theoretical metrics but also closely related to the deployed measurement equipment. Specifically, we employed a commercially available 4-channel optical power meter (Keysight N7745A) with its detailed specifications presented in [12]. During the experimental testing, the averaging time was configured to 1 μ s and the power meter range was set to [-40, 0] dBm, to comply with the dynamic range of the optical power measured at the column outputs of the Xbar, after applying the fidelity restoration, i.e., [-34.4, -31.5] dBm. Such conditions, according to the power meter specifications, correspond to typical noise values of $\sigma_{noise_limit} = \sim 0.005\%$, which comprises a hard noise-limit for our experiment, as the theoretical-model derived limit is significantly lower due to the low input update rate utilized in this experimental study, resulting in a limited noise bandwidth and as such extremely low fundamental shot and relative intensity noise profiles. In section 2 of the main manuscript, the standard deviations of the programming error for the 4 Xbar columns were measured to be equal $\sigma_{equip} = [0.3 - 0.6]\%$, well above from the power meter -originating noise limit. As already discussed, this is expected, as the majority of the programming error originates from the environmental and mechanical perturbations during the experimental testing and can be compensated through appropriate packaging techniques [13],[14]. The packaging may include thermal management solutions such as heat sinks, thermal pads and other cooling mechanisms to regulate the temperature variation of the photonic chips. Additionally, mechanical stress and vibrations can be minimized by employing stable mounting and isolation mechanisms [13],[14]. We believe that by implementing these effective packaging strategies, it is possible to mitigate the impact of temperature variations and mechanical instability, and as such operate our Xbar prototype closer to its theoretical limits.

Finally, the range of the employed reverse bias voltages (Fig. 2) was implicitly defined through the breakdown voltage of the constituent EAM devices. More specifically, voltages lower than -3V could lead to higher risk of breakdown or damage to the device. In principle, alternative EAM designs could allow for higher voltage ranges and as such to a higher dynamic range.

Changes made in the text:

At the end of section 2, line 176, we added the following:

“The measurements across the 4 column outputs revealed standard deviations in the range of [0.014, 0.028] dB that correspond to $\sigma_{equip}=[0.3-0.6]\%$, while the noise limit of our experiment is dictated by the deployed measurement equipment. Specifically, we employed a commercially available 4-channel optical power meter (Keysight N7745A) with its specifications summarized in Σφάλμα! Το αρχείο προέλευσης της αναφοράς δεν βρέθηκε. and corresponding to a typical noise value of σ (noise_limit) $\approx\sim 0.005\%$, significantly lower than the measured error. Therefore, the majority of the programming error originates from the environmental and mechanical perturbations during the PIC measurement procedure Σφάλμα! Το αρχείο προέλευσης της αναφοράς δεν βρέθηκε. Σφάλμα! Το αρχείο προέλευσης της αναφοράς δεν βρέθηκε. By implementing effective packaging strategies, it is possible to mitigate the impact of temperature variations and mechanical instability, and as such operate our Xbar prototype closer to its theoretical noise limits.”

References for Reply Letter

- [1]. W. R. Clements et. al. "Optimal design for universal multiport interferometers," *Optica* 3, 1460-1465 (2016)
- [2]. P. L. Mennea, et. al., "Modular linear optical circuits," *Optica* 5, 1087-1090 (2018)
- [3]. G. H. Golub and C. F. Van Loan, "Matrix Computations," 4th ed. Baltimore, MD, USA: Johns Hopkins University Press, 2013
- [4]. S. Pai et al., "Parallel Programming of an Arbitrary Feedforward Photonic Network," in *IEEE Journal of Selected Topics in Quantum Electronics*, vol. 26, no. 5, pp. 1-13, Sept.-Oct. 2020
- [5]. R. Hamerly, S. Bandyopadhyay, and D. Englund, "Stability of self-configuring large multiport interferometers," *Physical Review Applied*, vol. 18, no. 2, 2022
- [6]. G. H. Golub and C. F. Van Loan, *Matrix Computations*, 4th ed. Baltimore, MD, USA: Johns Hopkins University Press, 2013
- [7]. L. Fan et. al., "Multidimensional Convolution Operation with Synthetic Frequency Dimensions in Photonics," *Phys. Rev. Applied*, 18, 034088 (2022)
- [8]. L. Fan et al., "Experimental realization of convolution processing in photonic synthetic frequency dimensions." *Sci. Adv.* 9, eadi4956(2023)
- [9]. A. Tsakyridis, et al., "Photonic neural networks and optics-informed deep learning fundamentals". *APL Photonics* 1 January 2024; 9 (1): 011102
- [10]. Y. Bai et al., "Photonic multiplexing techniques for neuromorphic computing," *Nanophotonics* 12(5), 795–817 (2023)
- [11]. G. Giamougiannis et al.. "Analog nanophotonic computing going practical: silicon photonic deep learning engines for tiled optical matrix multiplication with dynamic precision" *Nanophotonics*, vol. 12, no. 5, 2023, pp. 963-973. <https://doi.org/10.1515/nanoph-2022-0423>
- [12]. N7744A 4-channel optical multiport power meter N7745A 8-channel ... Available at: <https://www.keysight.com/us/en/assets/7018-01757/data-sheets/5989-7976.pdf>. (Accessed: 11th March 2024).
- [13]. I. -L. Bundalo et al., "PIXAPP Photonics Packaging Pilot Line – Development of a Silicon Photonic Optical Transceiver With Pluggable Fiber Connectivity," in *IEEE Journal of Selected Topics in Quantum Electronics*, vol. 28, no. 3: Hybrid Integration for Silicon Photonics, pp. 1-11, May-June 2022
- [14]. P. Karioja et al., "MEMS, MOEMS, RF-MEMS and photonics packaging based on LTCC technology," *Proceedings of the 5th Electronics System-integration Technology Conference (ESTC)*, Helsinki, Finland, 2014, pp. 1-6

REVIEWERS' COMMENTS

Reviewer #1 (Remarks to the Author):

I do not think that my concerns have been addressed. The novelty of the manuscript does not meet the requirement of this journal. So I have to suggest rejection of this manuscript.

Reviewer #2 (Remarks to the Author):

The authors have addressed my comments.